# The Inhibitory Effect of Endophyte-Infected Tall Fescue on White Clover Can Be Alleviated by *Glomus mosseae* Instead of Rhizobia

**DOI:** 10.3390/microorganisms9010109

**Published:** 2021-01-05

**Authors:** Jinming Liu, Xiaoyu Ge, Xiaowen Fan, Hui Liu, Yubao Gao, Anzhi Ren

**Affiliations:** College of Life Sciences, Nankai University, Tianjin 300071, China; 1120200490@mail.nankai.edu.cn (J.L.); xiaoyu@mail.nankai.edu.cn (X.G.); 2120190969@mail.nankai.edu.cn (X.F.); liuhui@mail.nankai.edu.cn (H.L.); ybgao@mail.nankai.edu.cn (Y.G.)

**Keywords:** endophytic fungi, *Glomus mosseae*, rhizobia, tall fescue, clover

## Abstract

In artificial ecosystems, mixed planting of gramineous and leguminous plants can have obvious advantages and is very common. Due to their improved growth performances and stress tolerance, endophyte-infected grasses are considered to be ideal plant species for grasslands. However, endophytic fungi can inhibit the growth of neighboring nonhost leguminous plants. In this study, we chose endophyte-infected and endophyte-free tall fescue (*Lolium arundinaceum* Darbyshire ex. Schreb.) and clover (*Trifolium repens*) as the experimental materials to explore whether arbuscular mycorrhizal fungi and rhizobium can alleviate the inhibitory effect of endophyte infection on clover. The results showed that endophytic fungi significantly reduced clover biomass. Arbuscular mycorrhizal fungi inoculation significantly increased the biomass of clover in both endophyte-infected tall fescue/clover and endophyte-free tall fescue/clover systems but the beneficial contribution of arbuscular mycorrhizal fungi was more obvious in the endophyte-infected tall fescue/clover system. Rhizobia inoculation could alleviate the detrimental effect of tall fescue on the growth of clover but did not alleviate the detrimental effect of endophyte infection on the growth of clover.

## 1. Introduction

Plant to plant interactions have important effects on the productivity and composition of plant communities and even ecosystems [1,2,3,4]. In artificial cultivation, the mixed planting method of leguminous and gramineous plants is usually adopted to build grassland. Compared with monoculture grassland, the mixed planting of grassland not only has advantages in improving forage yield and quality [5,6,7] but also plays an obvious role in improving soil fertility and realizing systematic sustainable production [8]. In a grazing experiment in Canada, Chen et al. [9] found that total amounts of N_2_ fixed in *Medicago sativa* and *Poa pratensis* mixed pastures were sufficient to replace N fertilizer and sustain plant protein for grazing compared with grass-only pastures. Li et al. [10] reported that in intensive farming systems, leguminous and gramineous mixed cropping played an important role in alleviating the inhibitory effect of N fertilization on nodulation and N_2_ fixation and the productivity of intercropping could also be improved. In addition to the fact that leguminous plants and rhizobium can form root nodules [11,12], terrestrial plants can also develop other nutrient absorption strategies with microorganisms; for example, higher plants associate with mycorrhizal fungi [13,14,15,16,17] and grass and endophytic fungi form symbionts [18,19,20].

Endophytic fungi are a kind of fungi that usually inhabit the aboveground tissues of healthy plants and do not cause obvious diseases [21]. Mutual relationships have been found between endophytes and some important cultivated grasses such as tall fescue and perennial ryegrass [22,23,24]. The host plants can provide endophytic fungi with photosynthetic products, nutrients and habitats; moreover, secondary metabolites produced by endophytic fungi can promote the growth and development of host plants [25] and improve the host’s resistance to biotic [26,27,28,29,30] and abiotic stresses [31,32,33] such as drought [34,35,36], nutrient deficiency [37], herbivory [38,39,40,41] and pathogens [25,42,43], Given the higher growth advantages of endophyte-infected (E+) over endophyte-free (E−) plants, E+ grasses are considered to be ideal plant species for artificial grasslands. However, endophyte infection could change the interspecific relationship between host grasses and neighboring plants [44]. Endophytes may affect the competitiveness of the host grass by changing their shoot and root growth [45,46,47,48], physiological response to abiotic stresses [49,50] or production of allelopathic substances [44,51,52,53]. For example, Hoveland et al. [54] and Malinowski et al. [55] found that when tall fescue was mixed with clover or alfalfa, endophytic fungi significantly inhibited the growth of leguminous plants, thus improving the competitiveness of host plants. Over a four year period, Clay and Holah [56] found that species richness declined and tall fescue dominance increased in infected plots relative to uninfected plots. Thus, as tall fescue has been widely planted for forage and turf, the potential effect of the endophyte on interspecific relationships and communities is large. Therefore, it is urgent to consider how to make full use of the benefits of endophytic fungi while reducing its adverse effects on other key species such as leguminous plants in the establishment of lawn. However, only a few relevant reports are available.

In grassland ecosystems, arbuscular mycorrhizal fungi (AMF) and rhizobia are also common. Their beneficial effects on host plants have been widely demonstrated [57,58,59,60,61,62,63]. AMF are obligate symbionts, which are fully reliant on the host plants to obtain carbon sources. They provide nutrients such as phosphorus to their hosts in turn and they can become parasitic under certain conditions such as high P availability. AMF members of the genus *Glomus* can form symbiotic associations with the roots of many grass species and legumes. Endophyte-infected grasses can also be infected with AMF simultaneously [64,65,66,67]. Studies show that endophyte infection can reduce the AMF colonization rates in tall fescue and perennial ryegrass [68,69,70]. Mack and Rudgers [65] further found that there was a negative correlation between the endophytic hyphae density in the leaf sheath of tall fescue and the colonization rate of AMF (*Glomus intraradices*). Liu et al. [69] found that the inoculation of mycorrhizal *Glomus* fungi reduced the concentration of endophytic fungi in ryegrass leaves and meristem. These studies indicate that the beneficial effects of endophytic fungi on the host grass can be offset by AMF. Clover is an important green manure and turfgrass that can form a natural symbiosis with rhizobia and establish a symbiotic system with AMF such as *Glomus mosseae* and *Glomus claroideum*, which confer complementary resources such as P and N to their host plants [71]. Considering that endophyte-infected grasses cannot form symbionts with rhizobia and the beneficial effects of endophyte infection are negatively affected by AMF, we hypothesize that AMF or rhizobia inoculation can offset the inhibitory effect of endophyte-infected tall fescue on leguminous plants such as clover.

In the present study, endophyte-infected tall fescue and clover were chosen as experimental materials. We aimed to answer the following questions: (1) Can the inhibition of endophyte infection on clover be alleviated by inoculating AMF and/or rhizobia? (2) What is the relationship between endophytic fungi, AMF and rhizobia?

## 2. Materials and Methods

### 2.1. Planting Site and Soil Sampling

Soil was collected from local grassland with nitrogen and phosphorus concentrations of 0.17% and 0.028%, respectively. The soil was homogenized and sieved (passed through a 0.5 cm mesh) and autoclaved at 1 atm pressure, 121 °C, for 20 min, three times with 24 h interval before use, to eliminate the interference of other microorganisms in the soil. Each pot (24 cm diameter, 18 cm depth) contained 5 kg of sterilized soil. Eight seedlings were planted in each pot. For the mixed planting mode, the planting density ratio of clover to tall fescue was set as 4:4. The experiment was conducted in a greenhouse with a 10:14 h light/dark photoperiod at Nankai University. The starting time was January 2019 and lasted for four months with water provided as needed during the experiment. All pots were randomized weekly to reduce position effects.

### 2.2. Experimental Design

The mixed planting experiment was designed as a three-factor random setup. Factor one was the endophytic fungal state of tall fescue set at two levels: infected (E+) and uninfected (E−). Factor two was the inoculation of AMF, also set at two levels: inoculation of AMF (AMF+) and non-inoculation of AMF (AMF−). Factor three was the inoculation status of rhizobia; the two inoculation levels were the inoculation of rhizobia (R+) and non-inoculation of rhizobia (R−). Five plant combinations and four inoculation regimes were included. The plant combinations were: (1) eight E+ tall fescue per pot, (2) eight E− tall fescue per pot, (3) eight clover per pot, (4) four E+ tall fescue and four clover per pot and (5) four E− tall fescue and four clover per pot. Each treatment was repeated four times for a total of 80 pots.

### 2.3. Plant Materials

In this experiment, tall fescue (*Lolium arundinaceum* Darbyshire ex. Schreb.) and clover (*Trifolium repens*) were selected as the experimental materials. The tall fescue (Kentucky-31) can form a symbiotic relationship with only one endophytic fungus *Epichloë coenophiala* [72] The E+ and E− tall fescue seeds used in this experiment were originally provided by Professor Keith Clay at Indiana University and were multiplied in the growth chamber of Nankai University. The clover seeds were purchased from Jiangsu Earth Seedling base, China. Seeds were surface-sterilized in 5% sodium hypochlorite solution for 8 min before use. The tall fescue seeds used in this experiment and the seedlings that were subsequently grown were confirmed by the aniline blue staining method that endophytic fungi were present in the E+ plants but not in the E− plants. The seeds of the E+ and E− tall fescue were soaked in 5% (*w*/*v*) NaOH overnight before boiling in aniline blue for 5–8 min. For the seedlings, the leaf sheath epidermis was ripped and stained for 3–5 min. Both were then observed under a microscope to examine the endophytic fungi. Every pot of plants in this experiment was planted from seeds.

### 2.4. Mycorrhizal Fungi and Rhizobia

We chose *Glomus mosseae* (Gm), which was a broadly suitable strain and could colonize the two plant species mentioned above, for AMF inoculation. It belongs to the family of Glomeraceae and represents a unique *Glomus* group. The strains were provided by the Institute of Plant Nutrition and Resources, Beijing Academy of Agriculture and Forestry and the inoculum of Gm was propagated on *Trifolium repens* for three months in a greenhouse. The inoculum consisted of spores and mycelium, colonized root fragments and sand. A total of 135 g inoculum was added to the sterilized background soil per pot; the control group received the same amount of autoclaved inoculum and 75 mL of non-autoclaved inoculum filtrate to ensure the consistency of the microflora.

We chose *Rhizobium leguminosarum* (purchased from Beijing Beina Chuanglian biotechnology company) for the rhizobia inoculation because this species was originally isolated from wild white clover root nodules and was shown to successfully infect the white clover species used in our experiment. Before use, the tested strains were activated on YMA plates and incubated at 28 °C for three days and then transferred to triangular bottles containing 150 mL of liquid culture medium for the shaking bed culture (150 r·min^−1^, 28 °C, 30 h). The bacterial suspension (> 10^9^ bacteria·mL^−1^) [73] was injected into the root of each seedling for the first three days (100 µL each time, once a day). Non-inoculated pots were injected with the same volume of sterile water.

### 2.5. Harvest and Measuring

The shoot and root dry weights of clover and tall fescue were determined separately after oven-drying at 80 °C for 48 h. The nodule numbers of clover were recorded before weighing.

A weighted subsample of fresh roots was randomly taken. The roots were cut into approximately 1 cm lengths and were then cleared in 10% (*w*/*v*) KOH and stained with 0.05% trypan blue. The AMF colonization rate was recorded referencing the method of Biermann and Linderman [74], using Equation (1):AMF colonization rate (%) = 100 × Σ (0 × root segment number + 10 × root segment number + 20 × root segment number + 100 × root segment number)/total root segment number(1)

### 2.6. Statistical Analyses

Variance was analyzed using SPSS 22.0 software. Three-way ANOVAs were used to analyze the biomass data as well as the nodule numbers, with the two-level factors set as ‘infected by endophytic fungi (E±)’ (endophytic-infected, endophytic-free), ‘inoculation of AMF (AMF±)’ (inoculated with AMF, inoculated without AMF) and ‘inoculation of rhizobia (R±)’ (inoculated with rhizobia, inoculated without rhizobia). Two-way ANOVAs were then conducted to analyze the mycorrhizal colonization rate data, the influence of the endophytic fungi and the inoculation of rhizobia and the interactions between them on the clover mycorrhizal colonization rate. The significance level for all tests was 0.05.

## 3. Results

### 3.1. Clover Nodule Number

Endophytic fungi, Gm and rhizobia had marked effects on the clover nodule number (Table 1). Endophyte infection reduced the nodule number of the neighboring clover (Figure 1A,D). Both Gm and rhizobia inoculation significantly increased the clover nodule number (Figure 1B,C).

### 3.2. Mycorrhizal Colonization Rate of Clover

Neither endophytic fungi nor rhizobia had a significant effect on the mycorrhizal colonization rate of clover and there was no interaction between the two factors (Table 1).

### 3.3. Biomass of Clover

Endophytic fungi significantly inhibited the growth of clover and Gm inoculation significantly increased the biomass of clover (Table 1, Figure 2A,B), though only the extent of increase was affected by endophyte infection. The higher extent of improvement occurred when clover was mixed with E+ tall fescue where the biomass of clover in AMF+ treatment was 4.77 times higher than in AMF− treatments (Figure 2D). When mixed with E−, the biomass of clover was 1.81 times higher in AMF+ treatment compared with AMF− treatment (Figure 2D). Rhizobia inoculation could enhance the biomass of clover but a significant effect occurred only when the clover was mixed with E− plants (Figure 2C,E).

### 3.4. Biomass of Tall Fescue

Endophytic fungi and Gm inoculation significantly affected the biomass of tall fescue (*p* < 0.001). Endophyte infection significantly increased the biomass of tall fescue (Figure 3A) while Gm inoculation significantly decreased the biomass of tall fescue (Figure 3B). The effect of rhizobia inoculation on the biomass of tall fescue was affected by endophyte infection and the beneficial effect of rhizobia inoculation occurred only in E+ plants (Figure 3C).

## 4. Discussion

### 4.1. AMF Inoculation Can Alleviate the Inhibition of Endophytic Fungi on Clover

In nature, grasses can establish multiple symbiotic relationships with endophytic fungi and AMF simultaneously [67,68,69,70]. To date, several studies have examined the interactions between endophytes and AMF and their combined impacts on host plants but have neglected their combined impacts on nonhost plants. For example, Guo et al. [75] found that the roots of endophyte-free tall fescue were colonized more limitedly than those of pearl millet seedlings by Gm and endophytic fungi infection caused a further reduction in the extent of colonization. Antunes et al. [76] conducted two experiments and explained that *Neotyphodiurn coenophialum* infection reduced AM fungal spore germination and colonization in tall fescue. This phenomenon also exists in other grass species; Omacini et al. [77] found that endophyte-infected *Lolium multiflorum* had a lower level of mycorrhizal colonization compared with non-endophyte infection. On the other hand, AMF inoculation reduced the density or concentration of endophytic fungi [65,69]. Regarding the effects on host grass growth, Müller [78] found that the effects of endophytes might be enhanced or counterbalanced in the presence of AMF in different grass-endophyte symbionts. Omacini et al. [74] and Mack and Rudgers [65] found that there was no significant interaction between beneficial endophytic fungi and neutral AMF on the growth of ryegrass or tall fescue.

In the present study, we found that the growth of tall fescue was improved by endophyte infection and inhibited by Gm inoculation; we also did not find significant interactions between endophytic fungi and Gm on the growth of the host tall fescue. However, regarding the neighboring plant clover, we found that its growth was inhibited by endophyte infection and improved by Gm inoculation and that the detrimental effect of endophyte infection could be alleviated by Gm inoculation. In a previous study, Wagg et al. [58] reported that AMF inoculation could alleviate the inhibitory effect of perennial ryegrass on clover. In the present study, we further found that AMF inoculation could alleviate the inhibitory effect of endophyte infection on clover.

### 4.2. Rhizobia Inoculation Can Alleviate the Detrimental Effect of Tall Fescue on the Growth of Clover but Not the Detrimental Effect of Endophyte Infection

Rhizobia are natural symbionts of leguminous plants and it is well known that the symbiosis is mutually beneficial. As endophyte-infected grasses are not infected by rhizobia, studies on the interaction between endophytes and rhizobia are limited. Omacini et al. [79] reported that rhizobia were sensitive to endophytic fungi. García Parisi et al. [80] found that endophytic fungi could inhibit the number of nodules in leguminous plants but did not affect their growth. Our study found that rhizobia inoculation could significantly increase clover biomass when mixed with E− tall fescue; however, when mixed with E+ plants, the beneficial effect of rhizobia inoculation on clover disappeared. That is, rhizobia inoculation could alleviate the detrimental effect of tall fescue on the growth of clover but did not alleviate the detrimental effect of endophyte infection on the growth of clover. The probable reason for this difference might be the rhizobia inoculation rate and, thus, its contribution was significantly inhibited by endophyte infection.

In conclusion, we found that the growth of clover was inhibited by endophyte infection and improved by Gm inoculation and that the detrimental effect of endophyte infection could be alleviated by Gm inoculation. Rhizobia inoculation could alleviate the detrimental effect of tall fescue on the growth of clover but did not alleviate the detrimental effect of endophyte infection on the growth of clover. Certainly, more AMF and rhizobia species should be tested to clearly explore the interaction among endophytes, AMF and rhizobia in affecting plant growth.

## Figures and Tables

**Figure 1 microorganisms-09-00109-f001:**
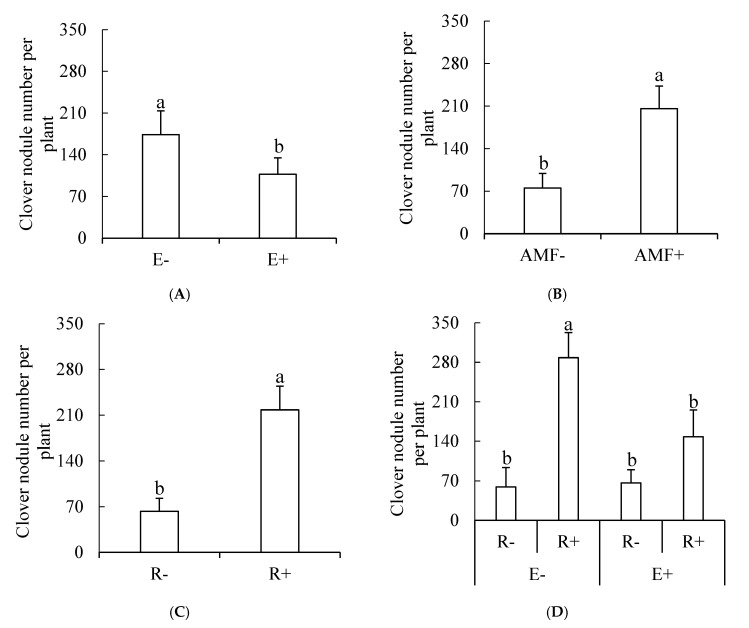
(**A**–**D**) The influence of endophytic fungi, AMF and rhizobia on the number of nodules. E+, endophyte-infected tall fescue; E−, endophyte-free tall fescue; AMF+, AMF-inoculated; AMF−, AMF-free; R+, rhizobia-inoculated; R−, rhizobia-free. Different small letters indicate significant differences between treatments (*p* < 0.05).

**Figure 2 microorganisms-09-00109-f002:**
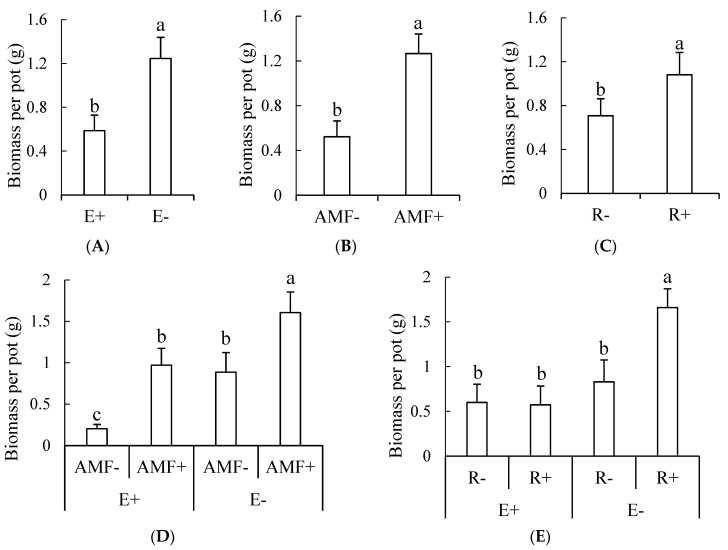
(**A**–**E**) The influence of endophytic fungi, AMF and rhizobia on the biomass of clover. E+, endophyte-infected tall fescue; E−, endophyte-free tall fescue; AMF+, AMF-inoculated; AMF−, AMF-free; R+, rhizobia-inoculated; R−, rhizobia-free. Different small letters indicate significant differences between treatments (*p* < 0.05).

**Figure 3 microorganisms-09-00109-f003:**
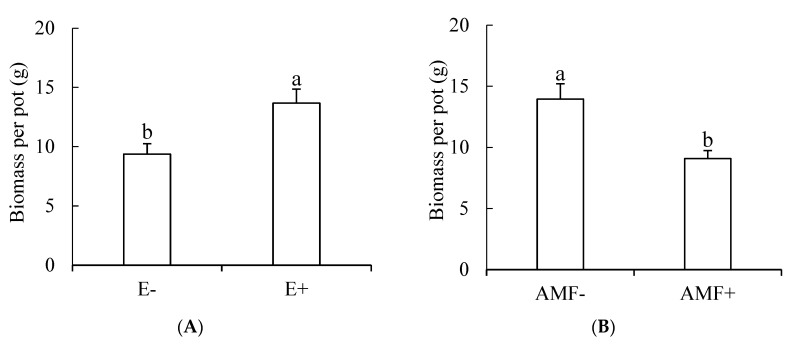
(**A**–**C**) The influence of endophytic fungi and rhizobia on the biomass of tall fescue. E−, endophyte-free tall fescue, E+, endophyte-infected tall fescue; AMF+, AMF-inoculated; AMF−, AMF-free; R+, rhizobia-inoculated; R−, rhizobia-free. Different small letters indicate significant differences between treatments (*p* < 0.05).

**Table 1 microorganisms-09-00109-t001:** ANOVA of the influence of endophytic fungi, arbuscular mycorrhizal fungi (AMF) and rhizobia on nodule number, mycorrhizal colonization rate (%) and biomass of *Trifolium repens.*

Variable	Nodule Number		Mycorrhizal Colonization Rate (%)		Biomass	
	*F*	*p*	*F*	*p*	*F*	*p*
Endophytic fungi (E)	4.704	0.040 *	0.419	0.530	24.892	<0.001 ***
AMF	18.089	<0.001 ***			28.531	<0.001 ***
Rhizobia (R)	25.524	<0.001 ***	0.040	0.844	4.364	0.048 *
E × AMF	0.471	0.499			5.238	0.032 *
E × R	5.764	0.024 *	0.720	0.414	5.329	0.005 **
AMF × R	1.780	0.195			5.615	0.027 *
E × AMF × R	0.022	0.883			2.005	0.944

* *p* < 0.05, ** *p* < 0.01, *** *p* < 0.001.

## Data Availability

This data Saved in Mendeley Data.

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
