# Peer review of "The Inhibitory Effect of Endophyte-Infected Tall Fescue on White Clover Can Be Alleviated by Glomus mosseae Instead of Rhizobia"

_microorganisms, 2021, doi:10.3390/microorganisms9010109_

Round 1
Reviewer 1 Report
Comments to authors
The authors report the relationship among endophytic fungi, AMF and rhizobia and consider a possibility of establishing a symbiotic system plant- AMF, which confer complementary resources, such as P and N, to their host plants.
Also, this study is a valuable contribution to the understanding of wood AMF and endophyte-infected plants interaction, the authors make a good job but other than a few comments listed below:
M&M
- i) 2.1. Planting site and soil sampling
Could you add information about lighting used for experiments?
- ii) 2.3 plant materials - How old were experimental plants? germlings or seedlings were used for the experiment?
iii) Line 104: endophytic fungus Epichloë coenophiala, not fungi
- iv) I also guess that other endophytic fungi could grow from tested plants. Did you check presence other fungal endophytes? If fungal species were more than one, how do you know it's the Epichloë coenophiala?
- v) 2.4 Mycorrhizal fungi and rhizobia: Could you add please a couple of words to justify such choice of AMF and rhizobia? I guess that other AMF and bacteria could be associated with tested plants.
Results
- i) Line 162: 3.2. Mycorrhizal colonization rate of clover– why Fig 1B characterize the absence of interaction between the two factors. Probably Fig. 2?
- ii) Line 165-166 – I `d suggest use rate/extent instead degree
Discussion
My suggestion that discussion should be expanded using e.g. allelopathic potential Vázquez‐de‐Aldana, B. R., Romo, M., García‐Ciudad, A., Petisco, C., & García‐Criado, B. (2011). Infection with the fungal endophyte Epichloë festucae may alter the allelopathic potential of red fescue. Annals of applied biology, 159(2), 281-290. and Vázquez-de-Aldana, B. R., Zabalgogeazcoa, I., & García-Criado, B. (2013). An Epichloë endophyte affects the competitive ability of Festuca rubra against other grassland species. Plant and soil, 362(1-2), 201-213.
The effects of the endophyte on host growth under inter-specific competition could be variable. Could you discuss such statement that in the binary experiment of infected and non-infected plants (E+ and E-), the competitive ability of E+ plants could be related to higher plant health (E+) or greater E+ plant and their root growth or even allelopathic effect?
Author Response
We highly appreciate the valuable detailed comments or issues on our manuscript of ‘microorganisms-1049305’. The suggestions are very helpful to us and we have incorporated them in the revised paper.
I would also like to address these issues point-by-point as follows. And we hope you will be satisfied with our responses to the ‘comments’ and the revisions of the manuscript.
Response to Reviewer 1 Comments
Point 1: i) 2.1. Planting site and soil sampling: Could you add information about lighting used for experiments?
Response 1: Thanks for the reviewer’s comments.
We have added the information about lighting used for experiments, the experiment was conducted in the greenhouse with a 10:14 h light/dark photoperiod (See Line 91 in the revised manuscript).
Point 2: ii) 2.3 plant materials - How old were experimental plants? germlings or seedlings were used for the experiment?
Response 2: Thanks for the reviewer’s comments.
First, I am very sorry for the unclearly description of experimental plants.
Second, we have re-explained the plant materials used in our experiment that every pot of plants in this experiment are planted from seeds (See Line 113-114 in the revised manuscript).
Point 3: iii) Endophytic fungus Epichloë coenophiala, not fungi
Response 3: Thanks for the reviewer’s comments.
We really appreciate that you point out our fault in such detail, and we have corrected it and checked the entire article carefully (See Line 107 in the revised manuscript).
Point 4: iv) I also guess that other endophytic fungi could grow from tested plants. Did you check presence other fungal endophytes? If fungal species were more than one, how do you know it's the Epichloë coenophiala?
Response 4: Thanks for the reviewer’s comments.
The tall fescue used in our experiment is the species called Kentucky-31, which are multiplied in our growth chamber, and only infected with one endophytic fungus Epichloë coenophiala (See Line 106-107 in the revised manuscript).
Point 5: v) 2.4 Mycorrhizal fungi and rhizobia: Could you add please a couple of words to justify such choice of AMF and rhizobia? I guess that other AMF and bacteria could be associated with tested plants.
Response 5: Thanks for the reviewer’s comments.
First, both Glomus mosseae (Gm) and Glomus intraradices (Gi) are a broadly suitable strains and can colonized tall fescue and clover. Second, this two strains were provided by the Institute of Plant Nutrition and Resources, Beijing Academy of Agriculture and Forestry, but the inoculum of Gm propagated on Trifolium repens for 3 months in the greenhouse are just harvest, so the activity is higher than Gi, thus we choose Gm as the AMF inoculum (See Line 116-119 in the revised manuscript).
As for Rhizobium leguminosarum that used in our experiment, it is isolated from the root nodule of wild white clover and through pre-experiment, we proved that it could successfully infected the white clover used in our experiment, thus we choose Rhizobium leguminosarum as the rhizobia inoculum (See Line 123-126 in the revised manuscript).
Point 6: vi)3.2. Mycorrhizal colonization rate of clover– why Fig 1B characterize the absence of interaction between the two factors. Probably Fig. 2?
Response 6: Thanks for the reviewer’s comments.
First, I am so sorry for the wrong expression of Fig 1B.
Second, we have removed the Fig 1 and changed the captions of other figures. At the beginning, I just want to display the phenomenon of AMF colonization in the clover root, it not related to result of ANOVA (See Line 166 in the revised manuscript).
Point 7: vii) I`d suggest use rate/extent instead degree.
Response 7: Thanks for the reviewer’s comments.
According to the reviewer’s comments, we have changed the “degree” into “extent”, as the latter one is more often used than the former (See Line 169 and 170 in the revised manuscript).
Point 8: Discussion
viii)My suggestion that discussion should be expanded using e.g. allelopathic potential Vázquez‐de‐Aldana, B. R., Romo, M., García‐Ciudad, A., Petisco, C., & García‐Criado, B. (2011). Infection with the fungal endophyte Epichloë festucae may alter the allelopathic potential of red fescue. Annals of applied biology, 159(2), 281-290. and Vázquez-de-Aldana, B. R., Zabalgogeazcoa, I., & García-Criado, B. (2013). An Epichloë endophyte affects the competitive ability of Festuca rubra against other grassland species. Plant and soil, 362(1-2), 201-213.
Response 8: Thanks for the reviewer’s comments.
We have expanded the references relevant to our discussion (See Line 207-213 in the revised manuscript).
Point 9: ix) The effects of the endophyte on host growth under inter-specific competition could be variable. Could you discuss such statement that in the binary experiment of infected and non-infected plants (E+ and E-), the competitive ability of E+ plants could be related to higher plant health (E+) or greater E+ plant and their root growth or even allelopathic effect?
Response 9: Thanks for the reviewer’s comments.
We have added related content in the revised manuscript (See Line 52-54 in the revised manuscript).
Reviewer 2 Report
The English in this manuscript is extremely poor. Authors need to correct English prior to submission for this or any other journal.
Author Response
Response to Reviewer 2 Comments
Point 1:The English in this manuscript is extremely poor. Authors need to correct English prior to submission for this or any other journal.
Response 1: Thanks for the reviewer’s comments.
We have carefully checked the English in the article.
Reviewer 3 Report
The inhibitory effect of endophyte-infected tall fescue on white clover can be alleviated by arbuscular mycorrhizal fungi instead of rhizobia
The authors investigate the effects of AMF and rhizobia on endophyte-infected and non-infected tall fescue and clover.
The article is interesting and the research questions are interesting.
Overall comment:
The study investigated the following species: Glomus mosseae, Rhizobium leguminosarum and two plant species. I would be careful about expanding these species interactions to all "AMF and rhizobium" interaction. This does not mean that the findings are not interesting and of importance, but make sure that the manuscript does not over reach by asserting all AMF and rhizobium interaction. Be specific when you discuss your results that it is only between these species.
Comments in general and by line:
Title: The title may be misleading. The authors tested only one AMF fungus, so there is no data to determine how "all" arbuscular mycorrhizal fungi would respond. If the authors achieved the same results with lots of AMF fungal species than I would agree, but it would be better to state this particular AMF fungus in the title.
line 46: " The host plants can provide endophytic fungal photosynthetic products,..."
Consider revising: The host plants can provide endophytic fungi with photosynthetic products,...
Line 72: Considering that enfophyte-infected grasses cannot form symbionts with rhizobia.....
Did the authors check the roots of the grasses to verify they did not get infected with the rhizobia?
Line 109: "confirmed by the aniline blue staining method..."
This needs a citation.
Lines 136:" Three-way Anovas was used...."
Lines 140:" Three-way Anovas were conducted...."
Please review for agreement. I believe "Three-way ANOVAs were...."
Line 147: (Fig. 2B, 2C).
Line 148: (Fig. 2A, 2D).
The Figures should be in order in which they are mentioned. B and A appear out of order.
Line 150: (shot
Consider revising shot to: imaged or captured
Lines 165-166: "though only the degree was affected by entophyte infection."
This was part of the sentence was confusing. Consider revising.
Lines 169-171: "These data indicated...."
This is a conclusion or interpretation and should be moved to the conclusions section.
Below Line 171: There is a B and C below? Are there images missing?
Line 179: "affected the biomass of tall fescue. (P<0.001).
Please remove the . after fescue.
Discussion:
This section ends abruptly. I would suggest a summary paragraph at the end.
Author Response
Response to Reviewer 3 Comments
Point 1: Overall comment:
The study investigated the following species: Glomus mosseae, Rhizobium leguminosarum and two plant species. I would be careful about expanding these species interactions to all "AMF and rhizobium" interaction. This does not mean that the findings are not interesting and of importance, but make sure that the manuscript does not over reach by asserting all AMF and rhizobium interaction. Be specific when you discuss your results that it is only between these species.
Response 1: Thanks for the reviewer’s comments
We have to confess that our experiment dose not over reach by asserting all AMF and rhizobium, so it will be better to discuss our results in terms of specific species. We have revised the relevant content in the revised manuscript.
Point 2: Comments in general and by line:
Title: The title may be misleading. The authors tested only one AMF fungus, so there is no data to determine how "all" arbuscular mycorrhizal fungi would respond. If the authors achieved the same results with lots of AMF fungal species than I would agree, but it would be better to state this particular AMF fungus in the title.
Response 2: Thanks for the reviewer’s comments.
We have revised the title (See Line 3-4 in the revised manuscript).
Point 3: line 46: " The host plants can provide endophytic fungal photosynthetic products,..."
Consider revising: The host plants can provide endophytic fungi with photosynthetic products,...
Response 3: Thanks for the reviewer’s comments.
I am so sorry for the wrong expression and I have corrected it (See Line 46 in the revised manuscript).
Point 4: Line 72: Considering that enfophyte-infected grasses cannot form symbionts with rhizobia.....
Did the authors check the roots of the grasses to verify they did not get infected with the rhizobia?
Response 4: Thanks for the reviewer’s comments.
In our experiment, we just checked the roots of tall fescue through phenotypic that they could not form nodules after inoculation with Rhizobium leguminosarum.
Point 5:Line 109: "confirmed by the aniline blue staining method..."
This needs a citation.
Response 5: Thanks for the reviewer’s comments.
We have added the citation of aniline blue staining method (See Line 114-117 in the revised manuscript).
Point 6:Lines 136:" Three-way Anovas was used...."
Lines 140:" Three-way Anovas were conducted...."
Please review for agreement. I believe "Three-way ANOVAs were...."
Response 6: Thanks for the reviewer’s comments.
We really appreciate that you point out our fault in such detail, and we have corrected it and checked the entire article carefully (See Line 146 in the revised manuscript).
Point 7:Line 147: (Fig. 2B, 2C).
Line 148: (Fig. 2A, 2D).
The Figures should be in order in which they are mentioned. B and A appear out of order.
Response 7: Thanks for the reviewer’s comments.
We have changed the figures order according to the letters (See Line 157 in the revised manuscript).
Point 8:Line 150: (shot
Consider revising shot to: imaged or captured
Response 8: Thanks for the reviewer’s comments.
I am so sorry for the wrong expression and we have removed the Fig 1 (in the original manuscript) and changed the captions of other figures.
Point 9:Lines 165-166: "though only the degree was affected by entophyte infection."
This was part of the sentence was confusing. Consider revising.
Response 9: Thanks for the reviewer’s comments.
We have changed “degree” into “extent of increase”, we want to express the fact that the increased biomass of clover due to Gm inoculation was more when mixed with E+ plant than E- plant (See Line 173-174 in the revised manuscript).
Point 10:Lines 169-171: "These data indicated...."
This is a conclusion or interpretation and should be moved to the conclusions section.
Response 10: Thanks for the reviewer’s comments.
We have rephrased the description of the results (See Line 177 in the revised manuscript).
Point 11:Below Line 171: There is a B and C below? Are there images missing?
Response 11: Thanks for the reviewer’s comments.
I am very sorry for the formatting errors in the manuscript, the B and C below are belong to the figure in the next page and I have corrected this error. (See Line 180 in the revised manuscript).
Point 12:Line 179: "affected the biomass of tall fescue. (P<0.001).
Please remove the. after fescue.
Response 12: Thanks for the reviewer’s comments.
Thank you for being so careful in pointing out this mistake, we have corrected it and checked the entire article carefully (See Line 187 in the revised manuscript).
Point 13:Discussion:
This section ends abruptly. I would suggest a summary paragraph at the end.
Response 13: Thanks for the reviewer’s comments.
We have added a summary paragraph at the end of the manuscript (See Line 245-249 in the revised manuscript).
Round 2
Reviewer 1 Report
Since you added Glomus mosseae to the title, please add info to Introduction and MM about this species (taxonomy, ecological niche and so on) and discuss why this particular species and if the same impact (alleviation of inhibitory effect) will be for use of other AMF
Add Glomus mosseae to Keywords
Author Response
Response to Reviewer 1 Comments
Point 1: Since you added Glomus mosseae to the title, please add info to Introduction and MM about this species (taxonomy, ecological niche and so on) and discuss why this particular species and if the same impact (alleviation of inhibitory effect) will be for use of other AMF
Response 1: Thanks for the reviewer’s comments.
First, we have added the related content into the revised manuscript (See Line 66-79, Line 124-128 and Line 253-254 in the revised manuscript).
Second, the inoculum of Gm propagated on Trifolium repens for 3 months in the greenhouse are just harvest, so the activity is higher than other species kept in our laboratory, thus we choose Gm as the AMF inoculum
Point 2: Add Glomus mosseae to Keywords
Response 2:Thanks for the reviewer’s comments.
We have added Glomus mosseae to keywords (See Line 25 in the revised manuscript).
Reviewer 3 Report
The authors have made the suggested changes, thank you very much. Below are some minor suggestions.
Specific line issues:
Issue: (word order) Line 54-55: or allelopathic substances production Consider revising to" or production of allelopathic substances"
Issue: (change of to at) Line 91: of Nankai University. Consider revising to " at Nankai University."
Issue: (verb needs to be in the past tense: are to were) Line 116: "... in this experiment are planted from seeds." Consider revising to".. in this experiment were planted from seeds."
Issue: (Sentence clarity for the reader) The addition of these sentences helped fully explain why the rhizobia species was used- thank you very much. However, the below sentence is just a suggestion to make those sentences concise and clear to the reader. Lines 126-129: Consider revising to the following: We chose Rhizobium leguminosarum (purchased from Beijing Beina Chuanglian biotechnology company) for the rhizobia inoculation because this species was originally isolated from wild white clover root nodules and was shown to successfully infect the white clover species used in our experiment.
Issue: (Remove on the one hand as its is redundant) Line 210: "For example, on one hand, Guo...." Consider revising: " For example, Guo..."
Issue: (End the previous sentence and start a new sentence for clarity) Line 217: "on the other hand, " Consider revising: " . On the other hand,"
Author Response
Response to Reviewer 3 Comments
The authors have made the suggested changes, thank you very much. Below are some minor suggestions.
Specific line issues:
Point 1: Line 54-55: or allelopathic substances production Consider revising to" or production of allelopathic substances"
Response 1: Thanks for the reviewer’s comments
We have revised it according to the reviewer’s comments (See Line 54-55 in the revised manuscript).
Point 2: Line 91: of Nankai University. Consider revising to " at Nankai University."
Response 2: Thanks for the reviewer’s comments.
We have revised the prepositions (See Line 96 in the revised manuscript).
Point 3: Issue: (verb needs to be in the past tense: are to were) Line 116: "... in this experiment are planted from seeds." Consider revising to".. in this experiment were planted from seeds."
Response 3: Thanks for the reviewer’s comments.
I am so sorry for the wrong expression and I have corrected this mistake (See Line 121 in the revised manuscript).
Point 4: Issue: (Sentence clarity for the reader) The addition of these sentences helped fully explain why the rhizobia species was used- thank you very much. However, the below sentence is just a suggestion to make those sentences concise and clear to the reader. Lines 126-129: Consider revising to the following: We chose Rhizobium leguminosarum (purchased from Beijing Beina Chuanglian biotechnology company) for the rhizobia inoculation because this species was originally isolated from wild white clover root nodules and was shown to successfully infect the white clover species used in our experiment
Response 4: Thanks for the reviewer’s comments.
We have revised the sentence according to the reviewer’s suggestion (See Line 132-135 in the revised manuscript).
Point 5:Issue: (Remove on the one hand as its is redundant) Line 210: "For example, on one hand, Guo...." Consider revising: " For example, Guo..."
Response 5: Thanks for the reviewer’s comments.
We have removed “on one hand” (See Line 216 in the revised manuscript).
Point 6:Issue: (End the previous sentence and start a new sentence for clarity) Line 217: "on the other hand, " Consider revising: " . On the other hand,"
Response 6: Thanks for the reviewer’s comments.
We really appreciate that you point out our fault in such detail, and we have corrected it and checked the entire article carefully (See Line 222 in the revised manuscript).